# Untreated Vestibular Schwannoma: Analysis of the Determinants of Growth

**DOI:** 10.3390/cancers16213718

**Published:** 2024-11-04

**Authors:** Cheng Yang, Daniel Alvarado, Pawan Kishore Ravindran, Max E. Keizer, Koos Hovinga, Martinus P. G. Broen, Henricus (Dirk) P. M. Kunst, Yasin Temel

**Affiliations:** 1Department of Neurosurgery, Maastricht University Medical Center, 6202 AZ Maastricht, The Netherlands; daniel_vacan@hotmail.com (D.A.);; 2Department of Neurology, Maastricht University Medical Center, 6202 AZ Maastricht, The Netherlands; 3Department of Ear, Nose and Throat, Maastricht University Medical Center, 6202 AZ Maastricht, The Netherlands; 4Academic Alliance for Skull Base Pathologies, Maastricht University Medical Center, 6202 AZ Maastricht, The Netherlands; 5Academic Alliance for Skull Base Pathologies, Radboud University Medical Center, 6525 GA Nijmegen, The Netherlands; 6Istanbul Atlas University, 34406 Istanbul, Turkey

**Keywords:** systematic review, meta-analysis, vestibular schwannoma, growth rate, untreated, factors, skull base

## Abstract

This study investigates predictors influencing the growth of untreated vestibular schwannomas, a common benign brain tumor. We aim to identify predictors of tumor growth, which is crucial for making informed treatment decisions. We reviewed numerous studies and analyzed variables such as age, gender, tumor size, location, symptoms, and MRI signal characteristics. The findings reveal that larger tumor size, extra-canalicular location, cystic components, and vestibular symptoms are associated with tumor growth. These insights can guide clinicians in identifying patients who may benefit from more aggressive monitoring or intervention, potentially improving outcomes for individuals with this condition. This research contributes to better understanding the variability in vestibular schwannoma (VS) growth and emphasizes the importance of personalized treatment strategies.

## 1. Introduction

Vestibular schwannoma (VS; acoustic neuroma) is the most common benign tumor in the adult cerebellopontine angle, accounting for over 80% of tumors in this area [1]. These tumors originate from Schwann cells of the vestibular branch of the vestibulocochlear nerve and cause ipsilateral sensorineural hearing loss in over 90% of patients, dizziness or imbalance in up to 61%, and asymmetric tinnitus in 55% [2]. VS is typically considered a tumor with a slow growth occurring in sporadic and genetic forms. The lifetime prevalence of the sporadic type is estimated to be 1 in 500 [3]. The treatment of VS depends on the size and symptoms of the tumor. Treatment options include conservative treatment (wait and scan), radiation therapy, and planned partial or (gross) total surgical resection alone or in combination with radiation therapy [1,4].

The growth rate of VS shows great variability among individuals, with certain tumors maintaining stability or even showing regression. Other tumors remain quiescent for years and then show growth, and others undergo rapid expansion, reaching rates as high as 25 mm/year [5,6,7]. Understanding the causes of tumor growth is essential in making clinical decisions. Previous studies have identified initial tumor size and location as predictors of growth [8,9,10].

Here, we reviewed recent data on the factors linked to growth in sporadic VS. We investigated a number of variables including age, gender, tumor size, location, symptoms, and MRI signal intensity.

## 2. Materials and Methods

The protocol for this review can be found in the PROSPERO online database of systematic reviews (ID: CRD42024511743).

### 2.1. Search Strategy

This systematic review was performed in accordance with the Preferred Reporting Items for Systematic Reviews and Meta-Analyses (PRISMA) guidelines. The PubMed, EMBASE, and Cochrane databases were searched using the following search input: (Vestibular schwannoma) OR (Acoustic neuroma) AND (growth). Articles were screened from 1 January 2000 up to 1 January 2024. This was accomplished by exporting the search results (i.e., the articles) into EndNote (Clarivate Analytics), after which duplicates were removed. Therefore, the articles were screened by examining the title and abstract one by one by two authors (C.Y. and D.A.). Articles with titles and abstracts conforming with our inclusion criteria were then analyzed in their entirety by the two aforementioned authors. Analyzing their full text, articles were included if they fit our inclusion criteria. The process was confirmed by one author (Y.T.). Disagreements between authors were resolved by reaching a consensus.

### 2.2. Study Selection

The inclusion and exclusion criteria were defined prior to the search. Studies were included if they (1) were peer-reviewed original articles about patients with untreated VS, (2) reported the growth rate of the untreated VS, and (3) were written in English. Studies were excluded if they were (1) literature or systematic reviews, case reports, comments, books, information pages, animal, or phantom studies, (2) histological studies, (3) or written in a non-English language.

### 2.3. Data Collection and Quality Assessment

Data were collected from the included articles by two authors (C.Y. and D.A.F.) and confirmed independently by an additional author (Y.T.). The extracted data included authors, number of participants, sex, mean age, tumor side, location, cystic aspect of the tumor, symptoms, and tumor size. Missing data were defined as data that were not provided in the selected articles. In the case that patients were double reported in different publications, the publication with the largest data set was used for the analysis.

We assessed the quality of included studies based on the Newcastle–Ottawa Scale (NOS) for the quality of cohort studies [11]. Eight questions were assessed, and each satisfactory answer received 1 point, resulting in a maximum score of 9. Only studies for which the majority of the questions were deemed satisfactory (i.e., with a score of 7 or higher) were considered to be of high quality. Two authors (C.Y. and D.A.F.) independently evaluated the quality of each study. A third author (Y.T.) was designated to make a final decision if the initial two reviewers were unable to reach a consensus.

### 2.4. Statistical Analysis

Statistical analysis was performed using SPSS V.28 (IBM SPSS Statistics, IBM Corp, Armonk, New York, NY, USA). This platform was used to perform descriptive and inferential statistics on the accumulated data. The primary outcome for this review was to find the factors associated with growth in untreated sporadic VS. After applying inclusion and exclusion criteria and subsequently extracting the data, the decision and feasibility of conducting a meta-analysis were evaluated. R studio was used to analyze the odds ratio (OR) of the predictive factors for VS growth. The inverse variance method was used to merge data for a random-effects meta-analysis. Data are presented as a forest plot.

## 3. Results

### 3.1. Literature Search

Our search generated 1526 articles, consisting of 1251 articles in PUBMED, 171 in EMBASE, and 104 in Cochrane. After automatic screening based on the inclusion and exclusion criteria, 396 references were excluded. An additional 1072 articles were excluded based on the title and abstracts being unrelated to VS growth. Furthermore, two articles were excluded because we were unable to obtain the full original text. Of the remaining 56 articles, 14 were excluded due to irrelevant content, and 1 was excluded since it reported on a previously published cohort. Ultimately, 41 articles were found to be suitable for the analysis, of which 21 defined tumor growth as linear extension greater than 2 mm, 9 articles defined tumor growth as linear extension greater than 1 mm (one article used both linear extension and volumetrics), 6 articles defined tumor growth as volume greater than 20%, and lastly, 6 articles used other parameters (see Figure 1).

This figure shows the flowchart of the literature search, with a total of 1526 articles searched from the PUBMED, Cochrane, and Embase databases. After removing duplicate articles and screening according to inclusion and exclusion criteria, 1130 articles were obtained. After examining the titles and abstracts of each result, according to the exclusion criteria, all animal studies, studies related to non-sporadic VS, and studies not related to VS growth were excluded. As a result, 1072 articles were excluded, as well as 2 articles that could not be read in full text. The remaining 56 articles were examined in detail, which resulted in the exclusion of another 14 articles as a result of insignificant information on tumor growth rate and one article reusing an already mentioned cohort. The remaining 41 articles were included.

### 3.2. Risk of Bias Assessment

All 41 included articles were evaluated. According to the scoring system, 26 studies were excellent research and 15 studies were of medium-quality research (see Appendix A).

### 3.3. Findings

Of the 41 included studies, 31 were retrospective cohort studies, whereas the remaining 10 were prospective cohort studies; 6 out of the 41 studies were multi-center (see Table 1).

Overall, four studies indicated that no factor could predict growth in their series. Only three articles [12,13,14] showed that age has a predictive effect on tumor growth, and no study showed that gender and tumor laterality have an impact on tumor growth. Furthermore, six articles [15,16,17,18,19,20] reported the growth of tumors in the first year (see Figure 2), five of which [15,16,18,19,20] indicated that the growth in the first year can predict growth in the following years. The studies focused on the location of the tumor ear (intra- and or extra-canalicular), the initial tumor size, and symptoms such as hearing loss, vertigo, and imbalance. In addition, posture swing tests [21] and MRI texture features [22] have newly emerged as predictive factors for tumor growth.

This figure shows the relationship between the occurrence of regrowing tumors and follow-up time, with the horizontal axis representing follow-up time and the vertical axis representing the percentage of tumors that have grown. There are a total of six articles describing specific quantities, with the dotted line indicating only the number of growing tumors in the first and the fifth years.

### 3.4. Overall Statistical Analysis

In order to minimize heterogeneity, we selected articles with the same definition of tumor growth. Among the 21 articles that defined tumor growth as linear ≥2 mm, 5 articles [24,29,31,39,45] did not provide data that could be processed. Furthermore, one article [33] only included intra-canalicular tumors, two articles [22,40] limited growth to 2 mm per year, and one article [49] included both linear greater than 2 mm and volume greater than 20% in the growth group. Therefore, a total of 12 articles were included in the analysis.

A total of 6168 patients were included in the analysis, of which 2337 patients (37.8%) were in the growth group and 3831 patients (62.1%) were in the non-growth group, with growth patients accounting for approximately 37.9% of the total patients. The age range of the entire cohort was 16.5–88 years old, and the follow-up time range was 0.5–37 years. Gender, tumor side, hearing loss, and tinnitus were not predictive of tumor growth, while location, initial tumor size, cystic component existence, and vestibular symptoms can predict tumor growth, as described below (see Figure 3 and Figure 4).

This figure shows a forest plot that analyzes gender, tumor side, hearing loss, and tinnitus as risk factors with no statistical significance, where the study represents the study number. Experimental represents the growth group. Control represents the non-growth group. Events represents the number of males, tumor growth in the left, and the number of patients with hearing loss and tinnitus symptoms. OR is the odds ratio, and 95% CI is the 95% confidence interval, calculated using a random effects model and the inverse variance method. This figure also shows heterogeneity between groups and the variance *p*-value.

This figure shows a forest plot that analyzes vestibular symptoms, cystic components, location, and tumor size as risk factors with statistical significance, where the study represents study number. Experimental represents the growth group. Control represents the non-growth group. Events represents the number of patients with vestibular symptoms, the number with intra-canalicular tumors, the number of tumors with cystic components, and the number of tumors more than 10 mm and 15 mm in size. OR is the odds ratio and 95% CI is the confidence interval, using a random effects model and the inverse variance method. The figure also shows heterogeneity between groups and the variance *p*-value.

### 3.5. Patient Characteristics

In the study group, the male gender has a slight predominance, with an overall male-to-female ratio of 51:49. Among them, 656 males (50%, n = 1312) were in the growth group, and 940 males (51.8%, n = 1815) were in the non-growth group. There was no significant difference in tumor growth between the two sexes (*p* = 0.307).

### 3.6. Tumor Characteristics

The tumor side generally had a slight prevalence to the right side, with a ratio of 48 to 52. There were 79 cases (49.4%, n = 160) on the left side in the growth group and 137 cases (46.6%, n = 294) on the left side in the non-growth group. The laterality of the tumor was not statistically significant for tumor growth (*p* = 0.582).

In terms of the tumor component, reported in 1335 patients, the solid tumors have a predominance (89%). Among them, 89 cases (12.9%, n = 688) were cystic in the growth group, and 57 cases (8.8%, n = 647) were cystic in the non-growth group. The difference was statistically significant (*p* = 0.021), indicating that a tumor with cystic components is a risk factor for tumor growth.

The location of the tumor is divided into two groups, intra-canalicular (IC, 47.4%, n = 2523) and extra-canalicular (EC, 52.6%, n = 2799), with 715 IC (36.5%, n = 1959) tumors in the growth group and 1808 IC (53.8%, n = 3363) tumors in the non-growth group. There was statistical significance (*p* < 0.0001) indicating that at the initial diagnosis, tumor growth is more likely to occur in the EC than completely IC located tumors.

The initial tumor size was analyzed with two cut-offs, namely 10 mm and 15 mm. The 10 mm cut-off group included 1189 cases. Using this cut-off, 193 cases (52%, n = 371) had an initial tumor size smaller than 10 mm in the growth group, and 566 cases (69.2%, n = 818) were smaller than 10 mm in the non-growth group. There were a total of 1207 cases included in the 15 mm group. Among them, 334 cases (75.2%, n = 444) had an initial tumor size less than 15 mm in the growth group, while 638 cases (83.6%, n = 763) were found to be less than 15 mm in the non-growth group. The analyses of both cut-offs were statistically significant (*p* = 0.008, *p* = 0.002), indicating that at initial diagnosis, the larger the tumor is, the greater the likelihood of tumor growth.

### 3.7. Symptom Characteristics

The most common clinical symptoms were due to cochlear nerve involvement. As a result, 1960 cases (72.6%, n = 2698) presented with ipsilateral sensorineural hearing loss (SNHL) and 1243 cases (56.8%, n = 2190) with tinnitus followed by vertigo and imbalance caused by the vestibular nerve involvement (40.5%, n = 2673). Among the patients, there were 936 cases (75.5%, n = 1240) with hearing loss, 624 cases (61.4%, n = 1017) of tinnitus, and 549 cases (44.5%) of vestibular symptoms in the growth group. There were 1024 patients (70.2%, n = 1458) with hearing loss, 619 patients (52.8%, n = 1173) with tinnitus, and 533 patients (37.1%, n = 1438) with vestibular symptoms in the non-growth group. Among them, hearing loss and tinnitus were not statistically significant (*p* = 0.998, *p* = 0.067), while vestibular symptoms were statistically significant for tumor growth (*p* < 0.0001). The least-affected cranial nerve is the facial nerve.

## 4. Discussion

Our main goal was to identify and examine the factors that contribute to the growth of untreated VS. Our research identified several predictors of VS growth, which are the location of the tumor [16,18,19,32,35,36,38,41,43,46,47], initial tumor size [16,17,18,19,35,46], presence of a cystic component [46,47], and the presence of clinical vestibular symptoms [17,19,35,36,43,47].

The EC tumor location seems to be a predictor for tumor growth. We found that tumor growth is more likely to happen in VS that is located in an EC location in comparison with that located in an IC location, with 63.5% of the EC tumors showing growth, in contrast with only 36,5% of the IC tumors growing in a population of 1959 tumor growth patients. This finding is in line with the current literature [18,21,26,31,36,46,47]. Notably, Reznitsky et al. [41] reported, using data from the Danish national database, that EC tumors grew more than the IC tumors, and during a follow-up that lasted 10 years, this difference increased even more in the EC group. A possible explanation is the limited space that the IC tumors have to expand.

The initial VS tumor size with linear measurement was also statistically relevant. We divided our population into two different groups due to the different data found in the literature, one with a cut-off of 10 mm and another of 15 mm. When considering the 10 mm cut-off, 69.2% of the cases were smaller than 10 mm in the non-growth group, while 52% of the cases were smaller than 10 mm in the growth group. When considering the 15 mm cut-off, 75,2% of the cases were found to be less than 15 mm in growth group; meanwhile, in the non-growth group, 83,6% presented less than 15 mm. Thus, our results suggest that if the tumor is larger at the initial diagnosis, then it is more likely to grow. These findings are in line with the current literature [16,17,20,24,26,33,34,35,37,38,46]. Notably, the study by Agrawal et al. [27] found similar results, but also suggested that for every 1 mm increase in the initial tumor size, the probability of tumor growth increased by 20%. Hentschel [47] and coworkers also showed that with increasing Koos grade, the likelihood of growth also increases. This association between initial tumor size and growth is important to take into account for the management of VS, knowing that larger tumors might need closer radiological monitoring or surgical treatment.

The presence of cystic components is another important growth risk factor, as confirmed by Jon Sei Kim et al. [46] and Hentschel et al. [47], which is linked to the sudden expansion of cystic components.

We also found that the presence of clinical vestibular symptoms is predictive of tumor growth. Hearing loss was not associated with tumor growth. A possible explanation is that most tumors originate from the vestibular part of the auditory nerve. The current literature [17,30,38,48] also confirms that balance symptoms can be a potential predictor of VS growth. Breivik et al. [29] showed that vestibular symptoms such as dizziness were statistically correlated with tumor growth. Interestingly, two research groups [28,47] used imbalance complaints to establish a model for predicting tumor growth.

There is some evidence that MRI signal intensity can predict tumor growth. Hiroyuki et al. [10] found that higher signal intensity on contrast-enhanced MRI was present in growing VS. Takashi and associates [22] demonstrated a significant correlation between the minimum MRI signal intensity obtained through inverse difference moment normalization (Idmn) and the growth velocity of VS. These findings are interesting and suggest that investigating MR characteristics of VS can reveal new parameters to predict VS growth. In contrast to previous findings, our review found no predictive value of age, gender, or the side of the tumor in VS. This demonstrates the complexity of VS development and the limitations of using demographic variables as standalone predictors.

The results of this study also provide potential implications for the management of growth of schwannomas in other sites. For example, extracranial schwannomas, especially facial nerve schwannomas, also require detailed evaluation for diagnosis and treatment planning. The study by Vrinceanu et al. [52] explored the histological features of extracranial schwannomas of the facial nerve and emphasized the need for accurate assessment in treatment planning. Although the study did not directly evaluate growth potential, the emphasis on tumor location echoes our findings on the effect of VS IC and EC location on growth. This suggests that when managing extracranial schwannomas, factors such as location and clinical symptoms may also be instructive for tumor growth risk, thereby helping to develop more personalized follow-up and intervention strategies. These findings have cross-site applicability for predicting schwannoma growth, further suggesting that evaluating specific tumor characteristics can support more precise disease management.

### Limitations

Our research was primarily limited by the heterogeneity of the different studies used for analysis. Although we used a consistent definition of growth to minimize heterogeneity, the difference in methods used to measure VS among the different selected articles presents a potential source of bias. The lack of universal guidelines in the follow-up of VS and the diversity in the population investigated is another limitation and may have contributed to the discrepancy in the results between some studies. However, the study that we present provides a substantial amount of statistically significant data that can be used to better understand factors linked to the growth of VS.

## 5. Conclusions

We have identified four predictors that correlate with tumor growth: extra-canalicular tumor location, larger initial tumor size, cystic tumor component, and the presence of vestibular symptoms. The posture swing test and MRI signal intensity have emerged as new predictive factors. In contrast to previous findings, our review found no predictive value of age, gender, or the side of the tumor in VS.

## Figures and Tables

**Figure 1 cancers-16-03718-f001:**
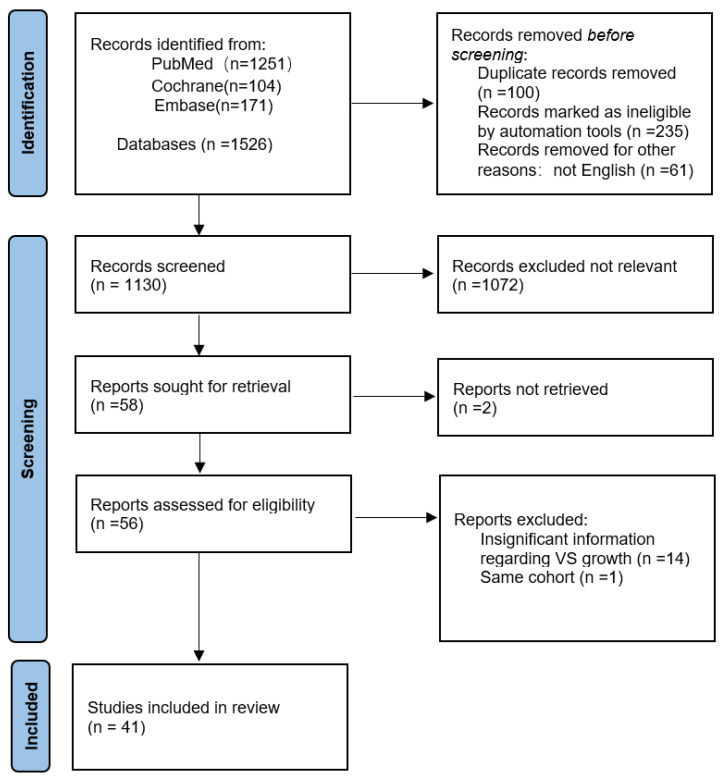
PRISMA 2020 diagram showing inclusion process.

**Figure 2 cancers-16-03718-f002:**
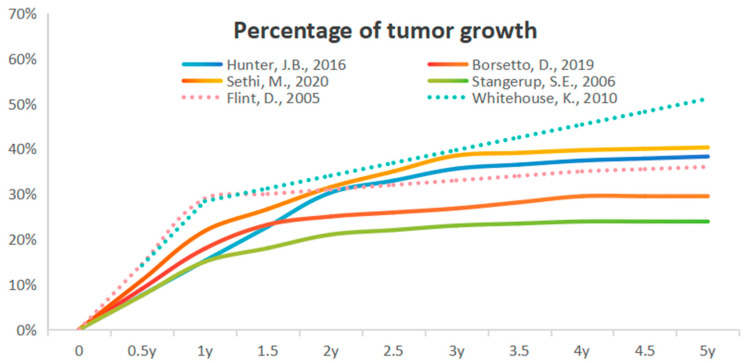
The percentage of tumor growth [15,16,17,18,19,20].

**Figure 3 cancers-16-03718-f003:**
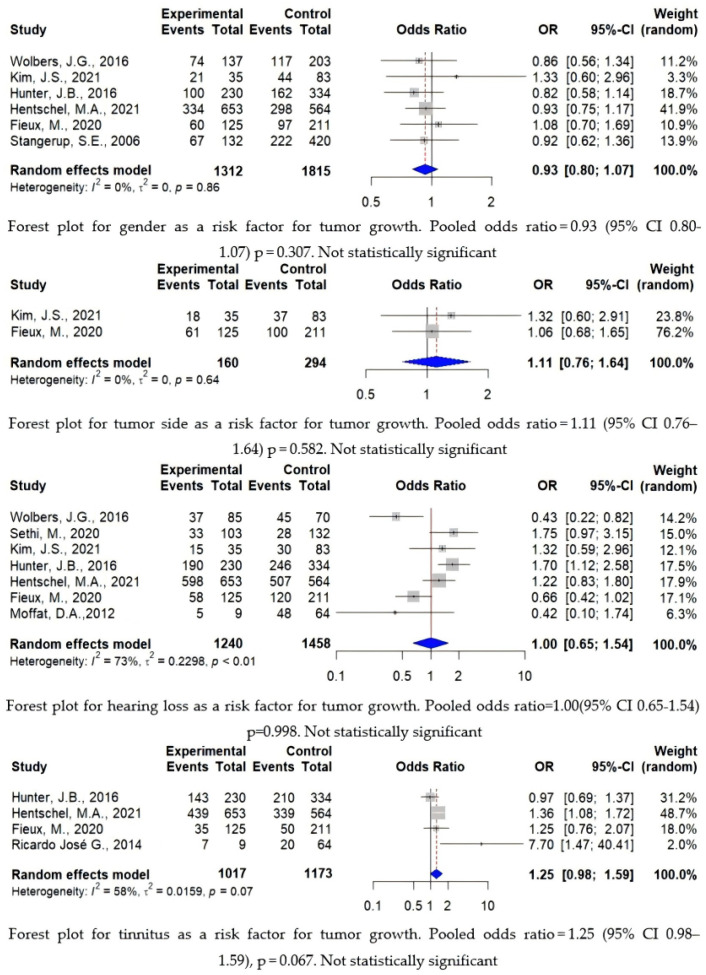
Gender, tumor side, hearing loss, and tinnitus as risk predictors with no statistical significance [16,17,19,32,35,36,43,46,47].

**Figure 4 cancers-16-03718-f004:**
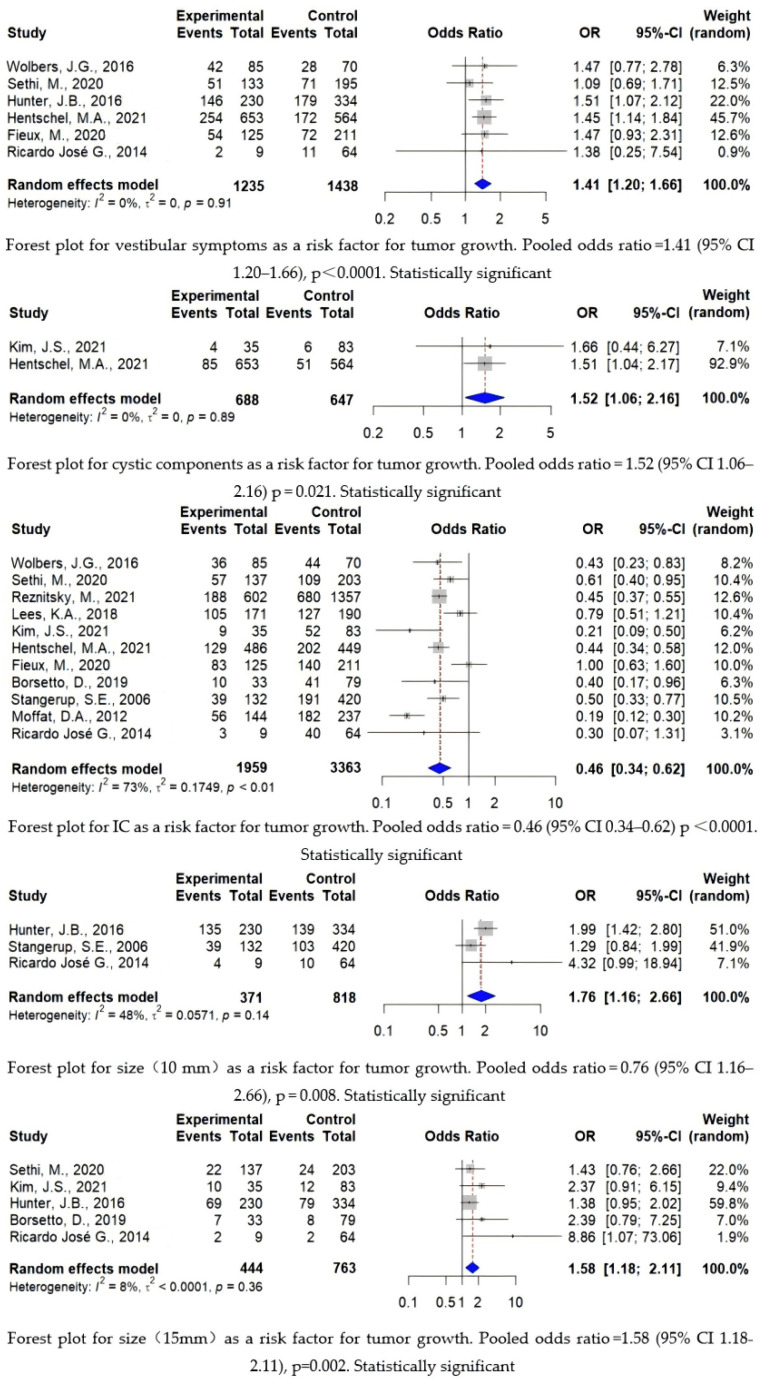
Vestibular symptoms, cystic components, location, and tumor size as risk predictors with statistical significance [16,17,18,19,32,35,36,38,41,43,46,47].

**Table 1 cancers-16-03718-t001:** General patient characteristics included in the study, as well as factors mentioned in the article that can predict tumor growth or not.

Study Number	Author, Year	Number of Patients			Association Factors	Non-Association Factors
		Total	Growth	N-Growth		
1	Diensthuber, M., 2005 [12]	118	n.m.	n.m.	Age, Location	n.m.
2	Prasad, S.C., 2018 [13]	576	n.m.	n.m.	Age	Location
3	Nilsen, K.S., 2020 [14]	204	n.m.	n.m.	Age, Imbalance	n.m.
4	Flint, D., 2005 [15]	100	36	64	First year	Age, Hearing Loss, Side, Location
5	Stangerup, S.E., 2006 [16]	552	132	420	1st year	Age, Gender
6	Hunter, J.B., 2016 [17]	564	230	334	Size, Imbalance	Age, Gender, Hearing Loss, Tinnitus, Vertigo
7	Borsetto, D., 2019 [18]	112	33	79	Location, 1st 1.5 years	Age, Hearing Loss
8	Sethi, M., 2020 [19]	340	137	203	1st year, Location, Size	Age, Gender, Hearing Loss, Imbalance
9	Whitehouse, K., 2010 [20]	88	45	43	1st year	Age, Size, Hearing Loss
10	Higuchi, Y., 2021 [21]	53	31	22	Location, Sway Velocity	Size, Age, Cystic, Hearing Loss
11	Itoyama, T., 2022 [22]	64	31	33	Min Signal, Idmn	Age, Size
12	Herwadker, A., 2005 [23]	50	n.m.	n.m.	n.m.	Age, Gender, Size, Side
13	Solares, C.A., 2008 [24]	110	23	87	Size (Women), Location	Age, Size (Men)
14	Bakkouri, W.E., 2009 [25]	325	n.m.	n.m.	Delay in Diagnosis	Location, Gender, Age, Hearing Loss, Koos Grade
15	Suryanarayanan, R., 2010 [26]	240	74	162	Location, Size	Age, Sex
16	Agrawal, Y., 2010 [27]	180	65	115	Size, Tinnitus	Age, Gender, Side, Location, Hearing Loss, Vertigo
17	Timmer, F.C., 2011 [28]	240	75	165	Location, Tinnitus Imbalance, Hearing Loss	Gender, Age, Side
18	Breivik, C.N., 2012 [29]	186	n.m.	n.m.	Tinnitus, Imbalance	n.m.
19	Jethanamest, D., 2015 [30]	94	n.m.	n.m.	Imbalance	n.m.
20	Hughes, M., 2011 [31]	59	n.m.	n.m.	Location	Age, Gender, Side, FN, Hearing Loss
21	Moffat, D.A., 2012 [32]	381	124	257	n.m.	Age, Size, Side, Gender
22	Lee, J.D., 2014 [33]	31	7	24	Size, Hearing Loss	n.m.
23	Tomita, Y., 2015 [34]	43	22	21	Size	Age, Gender, Location, Cystic
24	Ricardo José G., 2014 [35]	73	9	64	Location	n.m.
25	Wolbers, J.G., 2016 [36]	155	85	70	Location, Hearing Loss > 2 Years	Age, Gender
26	Daultrey, C.R., 2016 [37]	555	66	489	Location, Size	n.m.
27	Lees, K.A., 2018 [38]	361	172	189	Size, Imbalance, Aural Fullness	Age, Gender, Side, Hearing Loss, Tinnitus, Vertigo
28	D’Haese, S., 2019 [39]	62	35	27	n.m.	Age, Size, Location, Hearing Loss, Imbalance, Tinnitus
29	Kleijwegt, M., 2019 [40]	169	92	77	Hearing Loss, Cystic, Location	Age, Gender, Side
30	Reznitsky, M., 2021 [41]	1959	602	1357	Location	n.m.
31	Schnurman, Z., 2020 [42]	212	140	72	Size	Age
32	Fieux, M., 2020 [43]	336	125	211	Size, IAC Filling Stage, Hearing Loss *	Age, Gender, Location, Side
33	Marinelli, J.P., 2022 [44]	952	622	330	Size, Tumor Growth Rate	Age
34	Marinelli, J.P., 2021 [45]	592	357	235	Magnitude of Growth, Tumor Growth Rate	n.m.
35	Kim, J.S., 2021 [46]	118	35	83	Size, Location, Hearing Loss	Age, Gender, Cystic
36	Hentschel, M.A., 2021 [47]	1217	653	564	Imbalance, Tinnitus Koos Grade, Size	n.m.
37	Dardis, A., 2022 [48]	443	215	228	Imbalance, Size	n.m.
38	Truong, L.F., 2023 [49]	78	39	39	Hearing Loss < 2 years	Age, Gender, MRI Texture
39	Yagi, K., 2023 [50]	67	15	52	Hearing Thresholds at 1000 Hz	Age, Gender, Vertigo, Tinnitus, FN, Size, Location
40	Marinelli, J.P., 2023 [51]	405	n.m.	n.m.	n.m.	Age, Size
41	Yamada, H., 2022 [10]	31	15	16	Location, MRI Intensity	Cystic

* Means hearing loss associated with reduced risk of fast VS growth. This table displays the raw data of 41 included studies. It displays the number of patients included in each study, as well as the number of individuals in the growth and non-growth groups. In addition, this table portrays the variables associated with tumor growth and variables with no reported association.

## Data Availability

No new data were created.

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
