# Peer review of "Untreated Vestibular Schwannoma: Analysis of the Determinants of Growth"

_cancers, 2024, doi:10.3390/cancers16213718_

Round 1

Reviewer 1 Report

Comments and Suggestions for Authors

This systematic review conducted by Cheng Yang et al. turns out to be well elaborated in terms of study design with extensive analysis of the literature and major datasets available.

This study offers a precise evaluation of what predictors of growth are, and thus potential treatment, of vestibular schwannomas, providing an essential reference in clinical practice.

English is clear and fluent.

Literature is comprehensive and complete.

Text minor changes to be made:

Line 22: please, introduce full abbreviation description between parenthesis of VS before using it in the text e.g. “Vestibular Schwannoma (VS)”.

Lines 75 and 109: please, convert these line with no bold-writing.

Line 126: as an explanatory line, please decide whether to use full-text numbers or actual numbers (e.g. “twenty-six” or “26”).

Line 158: modify “one article 33”.

Line 168: “(see Figure. 3 and 4)” by removing the dot “.”

Line 180: modify “cystic exist” into “cystic components”.

Line 187: correct “than 10.15mm.”

Line 210: please, correct “namely 10mm” by spacing.

Line 280: “Limitations” is a sub-paragraph or a paragraph itself?

Author Response

We would like to thank you for your insights and feedback. Find below our point-by-point reply to your comments:

  1. Comment: Introduce the full abbreviation “Vestibular Schwannoma (VS)” when first mentioned in the text (Line 22).

  Response: We have updated Line 22 to introduce “Vestibular Schwannoma (VS)” when first mentioned, then consistently use the abbreviation “VS” thereafter.

  1. Comment: Remove bold formatting in Lines 75 and 109.

  Response: Lines 75 and 109 are in the regular format.

  1. Comment: Decide on a consistent format for number representation in Line 126, e.g., “twenty-six” or “26.”

  Response: We have standardized number representation throughout the manuscript, using Arabic numerals 26 for clarity, and you can find it in Line 134 now.

  1. Comment: Correct formatting errors (Lines 158, 168, 180, 187, 210).

  Response: We have corrected the noted formatting issues:

   - Line 158: changed “one article 33”to"one article [33]".

   - Line 168: Removed the dot after “Figure.”

   - Line 180: Changed “cystic exist” to “cystic components.”

   - Line 187: Corrected “than 10.15mm” to "than 10 mm and 15 mm"

   - Line 210: Inserted a space in “namely 10 mm” for readability.

and you can find it in Line 166,177,189,196,219 now

  1. Comment: Clarify whether “Limitations” is a sub-paragraph or a standalone paragraph (Line 280).

  Response: “Limitations” is a sub-paragraph in discussion section.

Reviewer 2 Report

Comments and Suggestions for Authors

This is an absolutely timely systematic review to help the neurosurgeon or ENT surgeon decide what advice to give to his or her patient when detecting vestibular schwannoma. The authors provide a good analysis of the limitations of the study. The study was primarily limited by the heterogeneity of the different studies available in the literature. The authors used a consistent definition of growth to minimize heterogeneous ity, still, the difference in methods used to measure VS among the different selected articles presents a significant source of bias.  The lack of universal guidelines in the follow-up of VS and the diversity in the population investigated is another limitation that have contributed to the discrepancy in the results between among studies. However, this study provides a substantial amount of statistically significant data that can be used to better understand factors linked to growth of VS.

Author Response

Thank you for reviewing and commenting on our paper. We appreciate your insights and hope that this article can further clarify the current understanding and management of VS.

Reviewer 3 Report

Comments and Suggestions for Authors

Esteemed Authors,

I read with great interest your manuscript about the growth patterns of vestibular schwannoma.

However, there are some aspects that require your attention.

Regarding the selection of the article that are included in the study. In one step you remove from the database more than 1000 articles. Please expand on the reasons of these manuscripts being removed. One example could be that many of those are animal studies. This will increase the reproducibility of your bibliograhic research.

In Table 1, format the second column without all capital letters. Also include in square brackets the reference numbers for these 41 studies. All these studies should be part of the reference list also for a quick check by any future reader of the study.

In Figure 3, you write Not Statistically Significant sometimes in Italics. Please correct and use the same font for every instance.

In Figure 3, you write Statistically Significant sometimes in Italics. Please correct and use the same font for every instance.

In the Discussion section you should also expand on the possible utility of these findings regarding the growth of schwannomas to other sites such as extracranial schwanoma. One reference to this could be the work by Vrinceanu D, Dumitru M, Popa-Cherecheanu M, Marinescu AN, Patrascu OM, Bobirca F. Extracranial Facial Nerve Schwannoma-Histological Surprise or Therapeutic Planning? Medicina (Kaunas). 2023 Jun 17;59(6):1167. doi: 10.3390/medicina59061167. PMID: 37374372; PMCID: PMC10302795.

There are many abbreviations in the text, please add a list of abbreviations at the end of the manuscript in order to increase the readability of the manuscript.

Format the references according to MDPI instructions for authors and update them according to the instructions of the reviewers.

Looking forward to receiving the improved version of your manuscript.

Author Response

We would like to thank you for your insights and feedback. Find below our point-by-point reply to your comments:

  1. Comment: Expand on the reasons for excluding more than 1000 articles during the study selection process.

  Response: We have added a detailed explanation in the "Literature Search" section, outlining the primary reasons for exclusion, such as animal studies, studies related to non-sporadic VS, and studies not related to VS growth. This addition enhances the transparency and reproducibility of our study's bibliographic research process. This alteration can be found between lines 127 to 133.

  1. Comment: In Table 1, format the second column without all capital letters. Also include in square brackets the reference numbers for these 41 studies.

  Response: We have reformatted Table 1 by adjusting the all-capitalized formatting and including reference numbers in square brackets. All studies referenced in this table have been verified and included in the reference list.

  1. Comment: Ensure consistent font use in Figures 3 and 4 for “Not Statistically Significant” and “Statistically Significant.”

  Response: We have corrected the font inconsistencies in Figures 3 and 4, ensuring uniform formatting throughout.

  1. Comment: Expand the discussion to explore the potential relevance of these findings for extracranial schwannomas.

  Response: We have added a paragraph in the Discussion section in line 289 to 301 that explores the relevance of our findings for extracranial schwannomas, citing Vrinceanu et al. (2023) as an example of managing extracranial facial nerve schwannomas and linking our study’s predictors of tumor growth.

  1. Comment: Add a list of abbreviations at the end of the manuscript to enhance readability.

  Response: We have included a list of abbreviations in the end of the manuscript section, line 430, covering all abbreviations used in the manuscript in order of appearance for reader convenience.

6.Comment: Format the references according to MDPI instructions for authors and update them according to the instructions of the reviewers.

Response: We have formatted all references according to the MDPI author guidelines. All updates requested by reviewers have been applied to the manuscript.